# Design of a Focusing Mechanism Actuated by Piezoelectric Ceramics for TMA Telescope

**DOI:** 10.3390/s23104610

**Published:** 2023-05-10

**Authors:** Ying Lu, Changzheng Chen, Junqing Zhu

**Affiliations:** 1Changchun Institute of Optics, Fine Mechanics and Physics, Chinese Academy of Sciences, Changchun 130033, China; 2University of Chinese Academy of Sciences, Beijing 100049, China

**Keywords:** focusing technology, piezoelectric ceramic actuator, flexible support

## Abstract

With the development of space telescopes towards high-resolution and intelligent imaging, the scale and complexity of the focal plane components of large-aperture, off-axis, three-mirror anastigmatic (TMA) optical systems are increasing. Traditional focal plane focusing technology reduces the system reliability and increases the scale and complexity. This paper proposes a three-degrees-of-freedom focusing system based on a folding mirror reflector, with a piezoelectric ceramic actuator as the driver. An environment-resistant flexible support was designed for the piezoelectric ceramic actuator through an integrated optimization analysis. The fundamental frequency of the large-aspect-ratio rectangular folding mirror reflector focusing mechanism was around 121.5 Hz. After testing, it was found to meet the requirements of the space mechanics environment. This system shows promise for application to other optical systems as an open-shelf product in the future.

## 1. Introduction

Space telescopes are subject to vibrations and shocks during launch, as well as orbital microgravity and thermal radiation, resulting in the slight deformation of structures and the degradation of the imaging performance. On the one hand, a deformation of the structure will cause surface deformations of each optical element, which in turn affects the wave aberration of the optical system, resulting in a decrease in the imaging quality. The current solution is to design each mirror assembly with a high stiffness and low thermal distortion and to ensure that reasonable thermal control measures are taken in orbit. Active deformable mirrors are used for the optical compensation of large-sized, high-weight mirrors [1,2]. On the other hand, a structural deformation will lead to the rigid displacement of the optical element, mainly including a position change and a tilt change. Position changes are usually compensated for by focusing, and tilt changes are generally compensated for by adopting greater tolerances in the optical design and a higher stiffness in the structural design. These measures cause a sacrifice in the optical performance and an increase in system weight. For large-aperture space telescopes, a six-degrees-of-freedom (DOF) mechanism was designed for correction [3,4]. However, there are few reports about the adjustment mechanism for TMA optical systems with a large aperture and super aspect ratios. Large-aperture TMA space telescopes are sensitive to environmental changes and vibrations and are more susceptible to structural deformations due to off-axis characteristics. In addition to designing a reasonable support structure, it is necessary to design a precise focusing mechanism to correct the changed focal plane.

As the requirements for space telescopes increase, intelligent imaging, a high light weight, and a high resolution have become inevitable research trends in the development of space optics, which puts forward new requirements for the optimal design of each component of a space telescope. A traditional focusing mechanism is composed of motors and transmission systems, mainly including worm and gear focusing mechanisms, a screw focusing mechanism, and a cam focusing mechanism. They provide a high accuracy and resolution, but at the cost of increasing the complexity and cost and reducing the reliability of the structure [5,6,7,8,9]. Some recent studies [10,11,12,13,14] have proposed improved focusing mechanisms, but they are still based on traditional focusing mechanisms and there are still movable motion pair, motion transformation, transmission error, friction, and wear problems. The focus mechanism proposed in this paper applies piezoelectric ceramics in the actuator and works with a flexible structure at the same time. There is no gap, no movable motion pair, no friction, and no transmission error, and the piezoelectric ceramic actuator provides a sufficient driving force and focus stroke and achieves micron-level precision. At the same time, the focusing mechanism has a three-degrees-of-freedom adjustment capability, which is unmatched by traditional focusing mechanisms. By adjusting the displacement and rotation of the mirror reflector, the defocus and line-of-sight jitter error caused by changes in the optical interval and tilt angle are compensated for to ensure the imaging quality of the telescope in orbit. This reduces the stiffness requirements of the optical–mechanical structure to a certain extent and is more conducive to the miniaturization and weight reduction of space telescopes. The piezoceramic actuators used here can only withstand a certain amount of force in the lateral direction. A flexible support structure was designed and optimized to improve the environmental adaptability [15,16,17].

In the rest of this paper, Section 2 describes the focusing method. Section 3 introduces the composition and working principles of the focusing mechanism and the performance characteristics of the piezoelectric ceramic actuator. Section 4 provides the surface figure analysis. Finally, Section 5 details a modal analysis and a dynamic mechanical vibration test of the focusing mechanism.

## 2. Focusing Method

This space telescope is still in the developmental stage. A schematic of the simplified optical system of the TMA space telescope studied in this paper is shown in Figure 1. The optical system consists of four parts: the primary mirror, the secondary mirror, the third mirror, and a rectangular folding mirror reflector. The weight of the telescope is about 400 kg.

The off-axis TMA space telescope is out of focus mainly due to the structural and thermal deformation of the optical system in the space environment. Structural and thermal deformations of the off-axis TMA space telescope will cause changes in the position and attitude of the optical mirrors, resulting in a change in the focal length of the system; thus, the ground scene cannot be clearly imaged in the focal plane. The asymmetry of the off-axis TMA optical system causes the deformation of the optical mirrors to change asymmetrically as well, so that the angle between the light incident on the focal plane and the optical axis also changes. As shown in Figure 2, the solid lines show the system state at the time of the optical design and the dashed lines show the system state after the structural and thermal deformation. The changed focus plane has not only an axial displacement against the original focal plane but also an angle displacement. Therefore, to focus an off-axis TMA space telescope, not only does the out-of-focus amount in the optical system need to be adjusted, but the angle change caused by the asymmetric deformation of the system also needs to be adjusted.

The commonly adopted focusing method for a space telescope includes lens focusing, focal plane focusing, and reflector focusing [18,19,20,21]. Lens focusing is achieved by changing the relative position of the secondary mirror in this optical system, i.e., by changing the optical interval between the secondary mirror and the primary mirror and between the secondary mirror and the third mirror, so that the focal plane imaging is optimized. In the off-axis TMA space telescope, the secondary mirror is in a special position. Compared to the primary mirror and the third mirror, the size is much smaller, and the distortion of the secondary mirror has the greatest impact on the image quality. Secondary mirror focusing has the following characteristics: Firstly, the movement of the secondary mirror will cause interval changes between the primary mirror and the third mirror at the same time, which means that the optical range will change by twice the amount of focusing, so the requirements for the precision and control of secondary mirror focusing are very high. Secondly, due to the structural characteristics of the TMA space telescope and the requirements of the installation position with the satellite platform, the finite element analysis shows that the impact and vibration response of the system is the greatest at the position of the secondary mirror, which will cause great damage to the operating performance of the focusing mechanism and seriously affect the normal work of the focusing mechanism.

Focal plane focusing is achieved by changing the relative position of the focal plane in the optical system. It only changes the relative position of the focal plane to the folding mirror reflector and does not cause a change in the structural position of the optical system, nor does it cause a change in the surface figure of the optical system. For the large-aperture TMA space telescope studied in this paper, the field of view is an important indicator of the space telescope, which determines the ground coverage width of the space telescope, and a longer imaging focal plane is required to obtain a larger field of view. Due to the limited length of a single-line-array CCD, the current solution is to stagger multiple-line-array CCDs to extend the focal plane. The focal plane dimensions of this telescope are 1000 mm × 8 mm. The weight of the focal plane is 15 kg. The large size of the focal plane places high demands on the stability and shape of the focal plane plate. This large-sized focal plane formed by CCD-interlaced splicing has a better quality, large electronics, and extensive heat generation. Heat dissipation is also a problem that should be considered. Therefore, the focal plane focusing method of adjusting the focal plane array’s position to coincide with the focus plane is unsuitable for this large-aperture TMA space telescope.

Folding mirror reflector focusing is realized by translating and tilting the folding mirror reflector to make the focus plane coincide with the focal plane array of the telescope. The surface of the folding mirror reflector is flat, and it folds the optical path in the optical system to make the telescope more compact. There are no complex electronic components. For long-focal-length systems, this translate-and-tilt focusing method makes focusing more accurate. However, the support structure of the folding mirror reflector and the focusing mechanism must share the same position, which will cause the support structure of the folding mirror reflector and the focusing mechanism to lose stability and affect the figure of the folding mirror surface. Thus, if a folding mirror reflector is used for focusing, the figure of the surface must be calculated. In summary, the folding mirror reflector focusing method was selected for this large-aperture TMA space telescope.

After a thermo-optical analysis, it was deduced that the focusing mechanism is required to have the ability of a 2 mm translation and a 0.03 mm tilt.

## 3. Mechanical Design

Previously, focusing was achieved by the direct actuation of the motor and transmission mechanism to achieve translational focusing. For off-axis TMA telescopes, tilting out of focus is achieved by using the thermoelastic deformations of the material. Under conditions of stiffness and strength, different structural forms or different materials are used at the two ends of the supporting backplate to produce different deformations on the upper and lower sides of the backplate and compensate for the out-of-focus tilt. This paper presents a folding mirror focusing mechanism driven by a piezoelectric ceramic actuator. The focusing mechanism is facilitated by three sets of piezoelectric ceramic actuators, which drive the translation and rotation of the folding mirror to achieve three degrees of freedom for the focus. To improve the reliability, a flexible structure was also designed to be used with it.

The design of the focusing mechanism is based on stringent performance requirements. The mechanical design model of the focusing mechanism is shown in Figure 3. It consists of a rectangular, large-aspect-ratio reflector assembly and three pairs of focusing assemblies. The reflector assembly consists of a lightweight reflective mirror, an Invar insert, a flexural support, and a baseplate. The focusing assembly consists of a piezoelectric ceramic actuator, a flexible structure, and some connectors.

A model of the flexible actuating assembly is shown in Figure 4. The piezoelectric ceramic actuator’s main structure and the flexible structure’s outer ring are connected and fixed to the mainframe of the telescope. The actuating rod is connected to the movable inner ring of the flexible structure, and the other end of the inner ring of the flexible structure is connected to the baseplate of the reflector. These same three flexible actuating assemblies are mounted on the three ends of the triangular baseplate. By controlling these three independent actuators, the small displacement of the actuating rod is transmitted to the mirror baseplate through the flexible structure, and a high-precision, three-degree-of-freedom positioning adjustment is realized. The principles and properties of the piezoelectric ceramic actuator and the design and optimization of the flexible structures are described in detail in the following sections.

Regarding the material selection of the mirror reflector focusing assembly, the stiffness and thermal expansion coefficient should be considered. The linear expansion coefficients of different materials at the connection should be similar or consistent. SiC is selected as the reflector material, which has excellent mechanical properties, such as a greater specific stiffness, a lower linear expansion coefficient, and a better thermal stability than Zerodur and Be. Invar inserts were selected as the transmission between the flexural support and the mirror. The baseplate is made of AlSiC. The flexible structure is made of TC4. The physical properties of the selected materials are listed in Table 1.

### 3.1. Piezoelectric Ceramic Actuator

Piezoelectric ceramics are made from modified lead zirconate titanate. Piezoelectric ceramics are deformed in an electric field, a phenomenon known as the inverse piezoelectric effect. They are anisotropic and will deform differently when the same electric field is applied to the three axes of the crystal. Piezoelectric ceramics have the advantages of a fast response, good controllability, a simple structure, and a high positioning accuracy, and they meet the requirements of high-performance drive components in the aerospace field [22,23,24,25]. Piezoelectric ceramic motors can achieve micron-level precision, a nanometer-level resolution, anti-magnetic interference, a high-frequency response, and a large output force. They are relatively mature and ideal micro-displacement drivers.

Due to the above characteristics of piezoelectric ceramic actuators, the N-216 piezoelectric ceramic actuator from Physik Instrumente (PI) in Karlsruhe, Germany, was selected as the driving motor of the focusing mechanism. Its driving force is up to 600 N, its holding force is up to 800 N, its stroke is 20 mm, and its resolution is 5 nm. The main materials are aluminum and stainless steel, and its weight is 1.25 kg. A view of the actuator structure and the technical data are shown in Figure 5a and Table 2, respectively. The technical data and figures are from the user manual, which can be found on their official website. PI piezoelectric actuators are composed of two types of piezoelectric ceramics with different deformation patterns: high-load stepper actuators and tangential actuators. A high-load stepper driver exhibits a vertical telescoping effect when an electric field is applied, which allows the tangential actuator to clamp or unclamp the actuator rod. Conversely, a tangential actuator can exhibit a twisting effect in the horizontal direction after an electric field is applied. When a tangential actuator clamps the actuator after the high-load stepper driver is telescoped, the piezoelectric ceramics in the tangential actuator can move the actuator from side to side under the action of the electric field in different directions. A piezoelectric ceramic actuator consists of multiple high-load stepper drivers and tangential actuator units. Adjacent units work in opposite states, with one group of units holding the actuator rod while the other is relaxed. The two groups of units work alternately to move the actuator in a directional manner. When the electric field is applied in reverse order, the actuator moves in the reverse direction. This is shown schematically in Figure 5b.

This piezo actuator does not require any additional mechanical coupling elements. The piezo drive acts directly on the runner. There are no other moving parts that limit the precision and reliability. Preloading the piezoelectric actuators against the runner ensures the self-locking of the drive when at rest and powered down. As a result, it does not consume any power, it does not heat up, and the position is kept mechanically stable. The motion of the piezoceramic actuator is based on crystalline effects and is not subject to any wear. Unlike other piezo motor principles, the coupling of the piezo actuators to the runner is not subject to sliding friction effects; the feed is achieved by the physical clamping and lifting of the actuators. Thus, it has a long lifetime and a high reliability.

However, the piezo actuator can only withstand a maximum lateral force of 20 N. To prevent the actuator from being damaged during launch or while working, we designed a flexible structure to work together with the actuator.

### 3.2. Design and Analysis of the Flexible Structure

Satellites experience harsh dynamic environments during launch. To improve the stability and reliability of the focusing mechanism, a flexible structure that looks like a diaphragm column was designed. This structure retains the axial degree of freedom and restricts the other five degrees of freedom. It not only satisfies the motion range of the focusing mechanism, but also protects the actuating rod of the actuator from being damaged by lateral force. A static analysis and finite element modal analysis of the structure were undertaken. At the same time, a hammer test was carried out on the processed model. The results show that the finite element model is consistent with the processed model.

A requirement for the flexible structure is that the structure can buffer the external load in the radial direction and has a large moving stroke in the axial direction. With reference to some flexural designs in space systems [26,27,28,29,30,31] shown in Figure 6a, some elongated curved slots were cut through the ring to create an inner ring that was attached to the outer ring only by way of these narrow flexures. These flexures isolate the inner ring from the outer ring, so a minute distortion of the outer ring will not be transferred to the inner ring. To make the flexure design more compliant in the direction of motion and eliminate axial stress in the arms, we made the narrow flexure arms symmetrical and folded them back to double the effective length, as shown in Figure 6b. Based on the symmetrically folded flexure described above, two diaphragm flexures were assembled parallel to each other by two rigid support cylinders, as shown in Figure 6c. The outer support is rigidly attached to the outer periphery of the diaphragm flexure, and the inner support is attached to the inner periphery of the diaphragm flexure. The outer ring is mounted on the mainframe of the space telescope, and the inner ring is not subjected to stress from the distortion of the outer ring. The two diaphragm flexures provide a highly precise motion in the axial direction with low or zero friction and a relatively large stroke. A photo of the prototype is shown in Figure 6d.

The geometric dimensions of the structure with two diaphragm flexures assembled in parallel include the inner ring radius $r_{1}$, the outer ring radius $r_{2}$, the diaphragm flexure thickness T, the structure height H, the number of arms N, the ratio of arm width to slit width R ($R_{a}$/$R_{s}$), and the number of arm folding times n. Considering the assembly relationship with the mirror assembly, it was determined that $r_{1}$ = 15 mm, $r_{2}$ = 50 mm, and H = 30 mm. According to the structure size and processing technology, it was determined that n = 4. The variables that need to be optimized are the thickness of the diaphragm flexure T, the number of arms N, and the width ratio R. To analyze the relationships between the changes in the parameters, combinations of parameters, and the stiffness of the structure in the axial and radial directions, we accurately evaluated all combinations of all factors at all levels. After the calculations, the optimal dimensions of the flexible structure in the above design were determined to be N = 9, T = 0.5, and R = 1/1. The axial stiffness meets the requirements of the focusing stroke and the radial stiffness is sufficient to protect the actuator.

The low-order mode of the space telescope focusing mechanism is an important indicator for investigating its dynamic stiffness. Preliminary judgments were made for the dynamic stiffness of the structure during the preliminary design phase. The natural frequency and mode shape can be identified by a modal analysis. According to the flexible structure model determined by finite element analysis, the first four vibration modes of the flexible structure are shown in Figure 7.

The impact test method was adopted to identify the fundamental frequency of the flexible structure. Four modal measuring points are arranged on the test model. The measuring and incentive points are shown in Figure 8, and the results of frequency response are shown in Figure 9. The first three modal parameters of the result are extracted. A comparison between the finite element analysis results and the impact test modal frequencies is shown in Table 3. The modes are relatively consistent. The second and third modes are mechanically the same (tilting of the inner ring against the outer). In an ideally symmetric structure, they would have the same frequency. As the symmetry is not ideal, the modes separate. The test modal frequencies are consistent with the finite element analysis results.

## 4. Surface Figure Analysis

After ground assembly, testing, and then launching into orbit, the mirror will be deformed due to weightlessness. Meanwhile, there are temperature gradients and deviations in the internal temperature control of the telescope. After in-orbit thermal control, the operating ambient temperature of the entire telescope will be in the range of 20 ± 2 °C. The mirror will also be deformed due to the temperature. The support structure of the folding mirror and the focusing mechanism share the same position, which will cause the support structure of the folding mirror and the focusing mechanism to become less stable, causing the folding mirror’s surface shape to change. Finally, when the folding mirror focusing mechanism enables the folding mirror assembly to adjust the pointing angle through the three-point actuators, the forced displacement of 0.03 mm will also lead to the deformation of the mirror. The surface figure is the deviation between the optical mirror surface and the ideal surface. The surface figure is one of the most important accuracy indicators in space telescope optical systems. The PV (peak-to-valley) and RMS (root mean square) values are important parameters for evaluating the surface figure.

In summary, a static analysis of the folding mirror assembly was performed, where the main analytical conditions were gravity in three directions (X, Y, and Z), a temperature variation of ±4 °C, and a forced displacement of 0.03 mm. The results of the surface figure and rigid displacements for these five working conditions are shown in Table 4. *d*x represents the rigid displacement in the X direction and *r*x represents the rotation around the X direction. The surface figure nephogram of the gravity in the X direction is shown in Figure 10a. The surface figure nephogram of the gravity in the Y direction is shown in Figure 10b. The surface figure nephogram of the gravity in the Z direction is shown in Figure 10c. The surface figure nephogram of the temperature variation of 4 °C is shown in Figure 10d. The surface figure nephogram of the forced displacement condition is shown in Figure 10e. The surface figure nephogram is a deformation nephogram, which shows the deformation of the whole surface with respect to the original position where it was not subjected to load.

It can be seen that the folding mirror assembly can maintain the surface figure within λ/50 RMS (λ = 632.8 nm) under the above working conditions, which meets the design requirements.

## 5. Simulation and Test

Due to the use of a flexible structure, the stiffness of the mirror focusing assembly will be greatly reduced, and it is necessary to verify whether the stiffness meets the requirements of use through vibration tests. In the initial design phase, a modal analysis of the focusing mirror assembly was carried out to verify its dynamic stiffness. The boundary was defined as the full constraint of the six degrees of freedom of the nodes at the connection position of the telescope mainframe, which is consistent with the actual installation conditions. As can be seen from Table 5, the fundamental frequencies of the reflector focusing assembly were all greater than 100 Hz and the first natural frequency was 121.50 Hz, which is greater than the given index.

A vibration test was conducted on the the processed mirror focusing assembly. A shaking table was used to conduct vibration tests on the mirror focusing assembly in three directions: X, Y, and Z. The sine vibration frequency range was between 5 and 100 Hz. Before and after the sinusoidal vibration, a sinusoidal scan at 0.2 g, 10–2000 Hz, and 4O ct/min was performed to determine the fundamental frequency of the assembly and the resonance point; the latter was determined to check whether the assembly changed after the sinusoidal vibration to determine the reliability of the assembly. The site of the mechanical environment test for the reflector focusing assembly is shown in Figure 11 and the vibration conditions are shown in Table 6.

The data obtained from the first frequency sweep test in the X direction are shown in Figure 12a; the sinusoidal vibration response curve of the reflector focusing assembly is shown in Figure 12b; and the data obtained from the second frequency sweep test are shown in Figure 12c.

The data obtained from the first frequency sweep test in the Y direction are shown in Figure 13a; the sinusoidal vibration response curve of the mirror focusing assembly is shown in Figure 13b; and the data obtained from the second frequency sweep test are shown in Figure 13c.

The data obtained from the first frequency sweep test in the Z direction are shown in Figure 14a; the sinusoidal vibration response curve of the mirror focusing assembly is shown in Figure 14b; and the data obtained from the second frequency sweep test are shown in Figure 14c.

From the figure, it can be seen that the fundamental frequency of the mirror focusing assembly was above 100 Hz, and there was no resonance point within 100 Hz. The sweep frequency curve after the sinusoidal vibration test was consistent with the first sweep frequency curve, which verifies that the stiffness of the mirror focusing mechanism fully meets the requirements of use.

## 6. Conclusions

Precision focusing mechanisms are a critical technology for the high-weight imaging of high-resolution and large-aperture TMA space telescopes. Simplifying the structure of the system, reducing the weight and cost, and achieving a better performance have always been the goals for space optical systems. In this paper, a piezoelectric-ceramic-actuated mechanism with a large aspect ratio mirror reflector greatly reduced the volume and weight of the focusing mechanism and improved its integration and reliability. After optimization and improvement, it could become a standard focusing mechanism, similar to open-shelf products that can be purchased by engineers and applied in various telescopes.

## 7. Patents

We obtained a Chinese invention patent for the focusing mechanism designed in this paper. The patent number is CN109270656B.

## Figures and Tables

**Figure 1 sensors-23-04610-f001:**
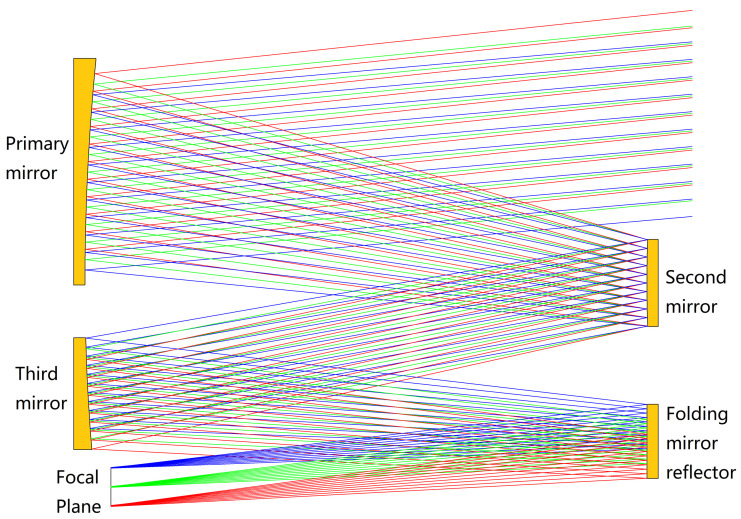
Simplified optical system schematic of the TMA space telescope. The line indicates the direction of light.

**Figure 2 sensors-23-04610-f002:**
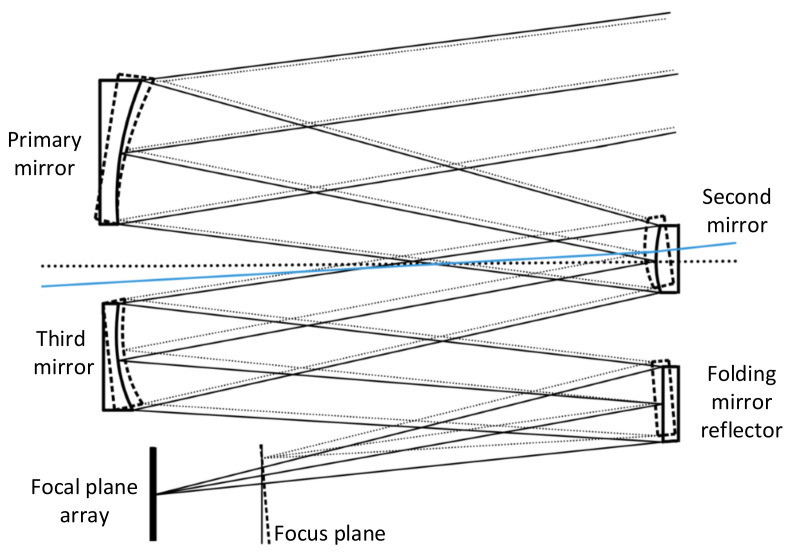
Schematic of the off-axis TMA optical system deformation and out-of-focus amount.

**Figure 3 sensors-23-04610-f003:**
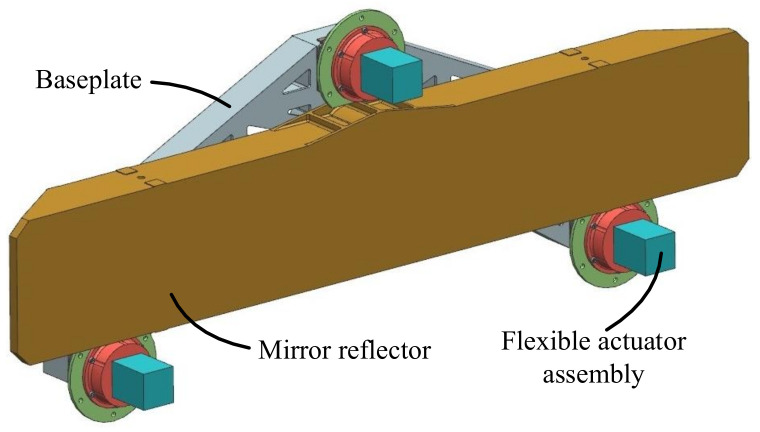
A mechanical model of the reflector focusing mechanism.

**Figure 4 sensors-23-04610-f004:**
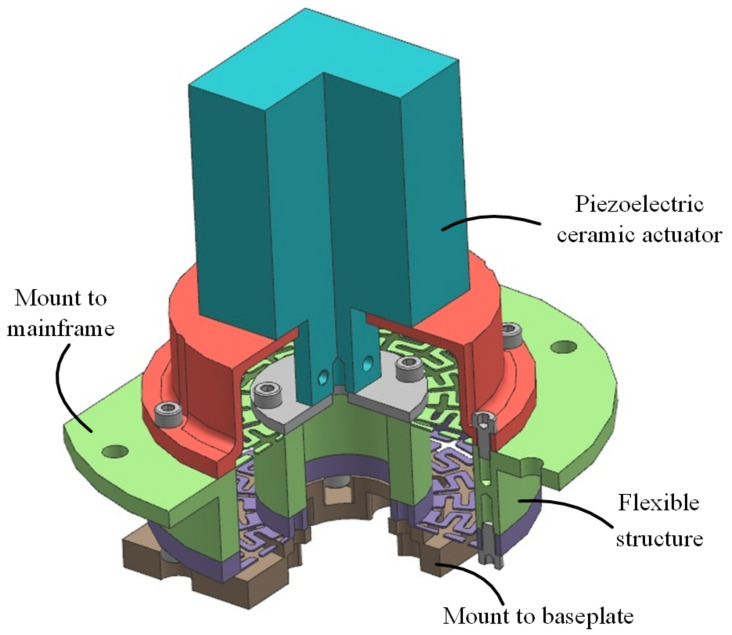
A mechanical model of the flexible actuating assembly.

**Figure 5 sensors-23-04610-f005:**
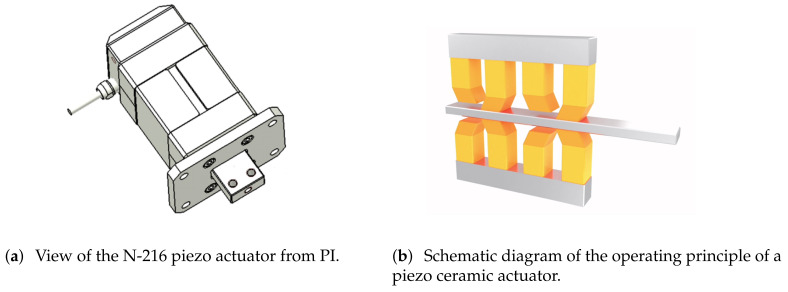
View of the N-216 piezo actuator and the operating principle of a piezo ceramic actuator.

**Figure 6 sensors-23-04610-f006:**
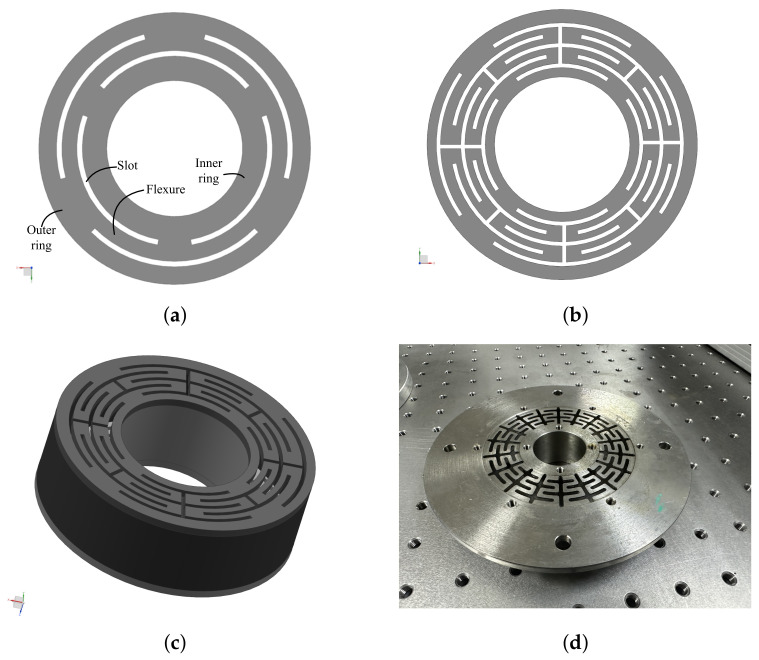
The design process for flexible structures. (**a**) A typical structure. (**b**) The initial design structure. (**c**) The final prototype of the flexible structure to be used. (**d**) A photo of the prototype.

**Figure 7 sensors-23-04610-f007:**
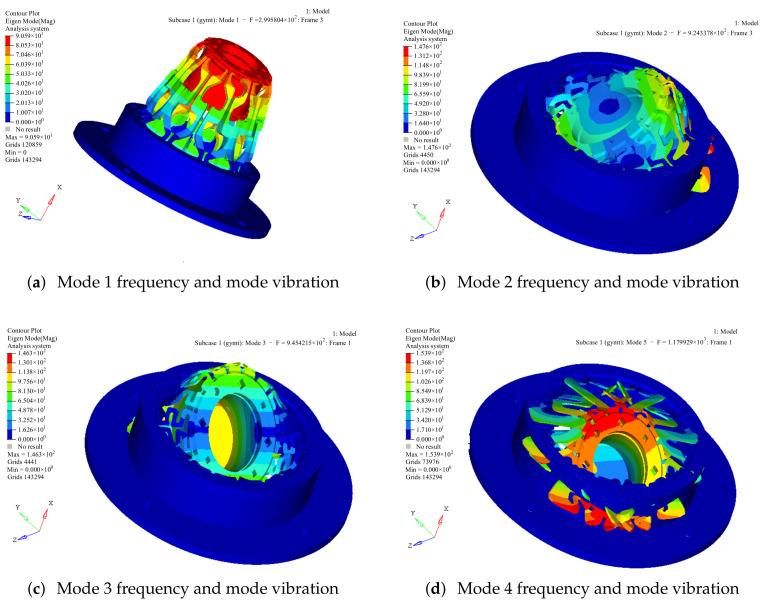
The first four modal frequencies and mode vibrations.

**Figure 8 sensors-23-04610-f008:**
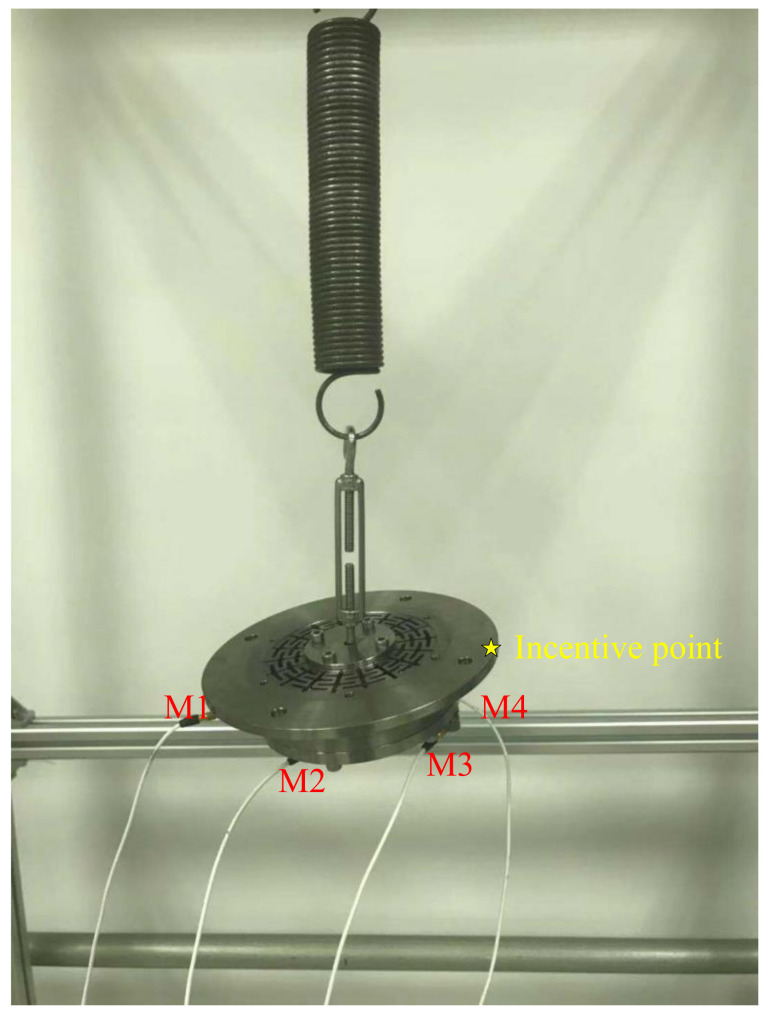
Positions of measurements and the incentive point, M1, M2, M3 and M4 are the measurement points.

**Figure 9 sensors-23-04610-f009:**
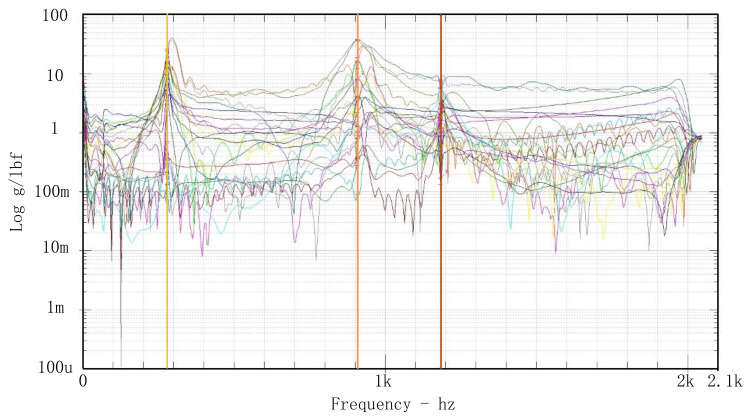
The results of frequency response curves.

**Figure 10 sensors-23-04610-f010:**
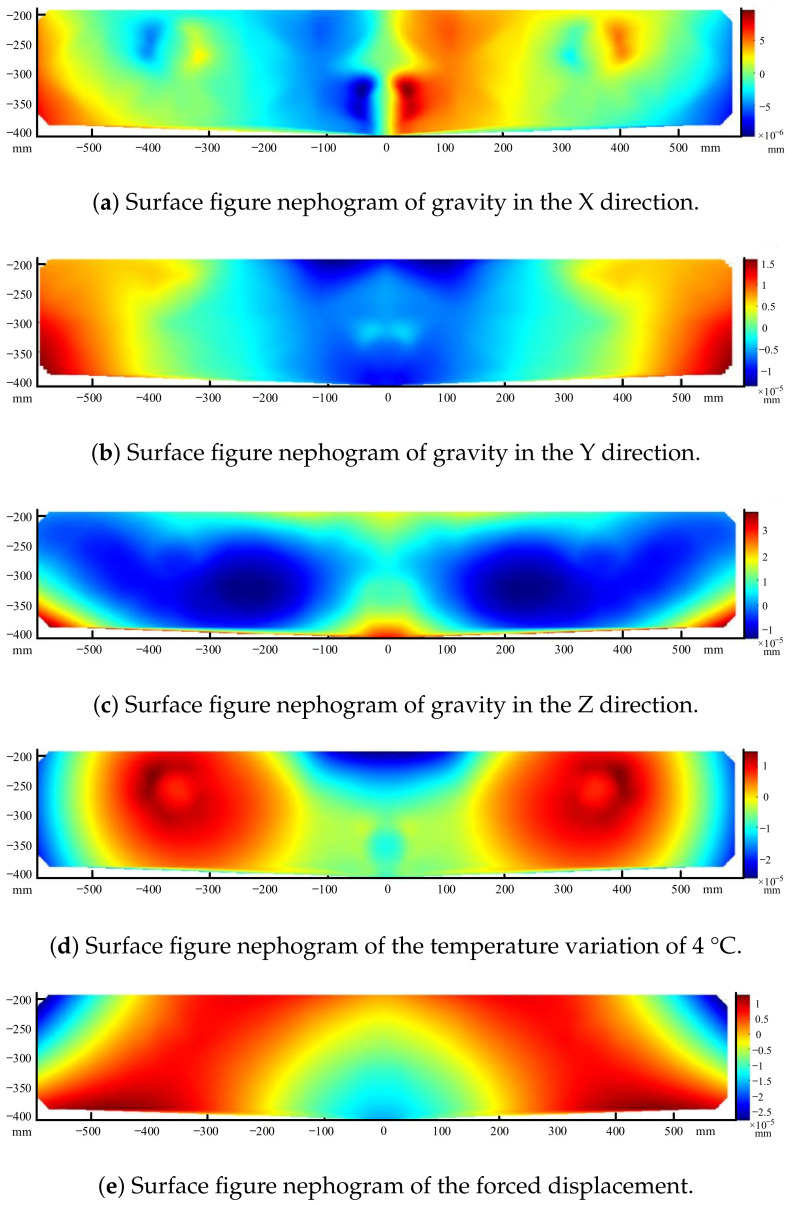
Surface figure nephograms of gravity, temperature, and forced displacement.

**Figure 11 sensors-23-04610-f011:**
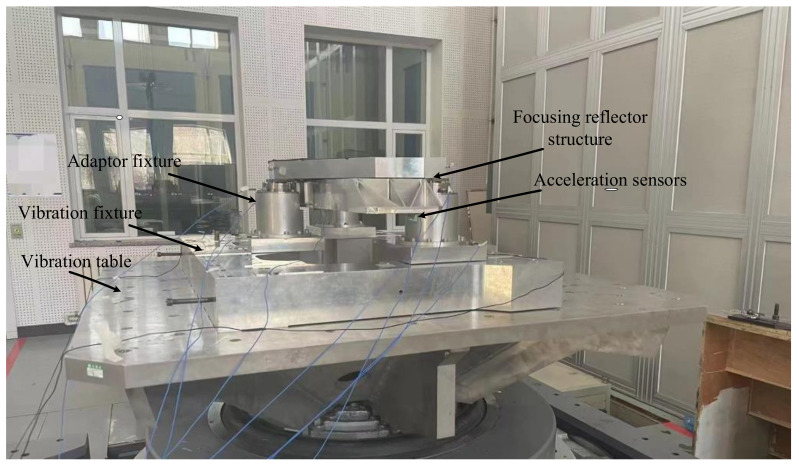
The site of the mechanical environment test.

**Figure 12 sensors-23-04610-f012:**
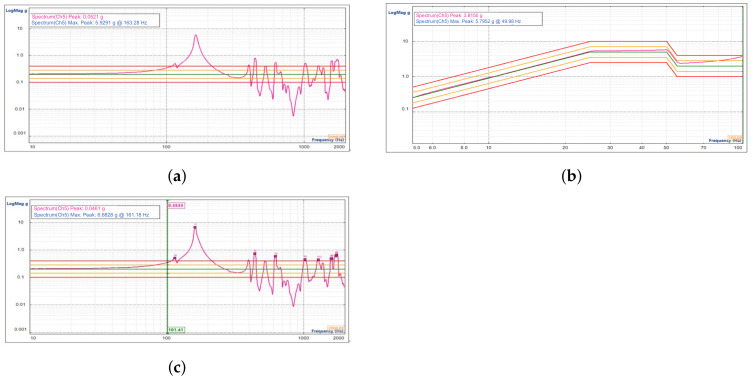
Results of the sinusoidal vibration test on the focusing reflector assembly in the X direction. (**a**) Results of the first sweep in the X direction. (**b**) Results of the sinusoidal vibration frequency response in the X direction. (**c**) Results of the second sweep in the X direction.

**Figure 13 sensors-23-04610-f013:**
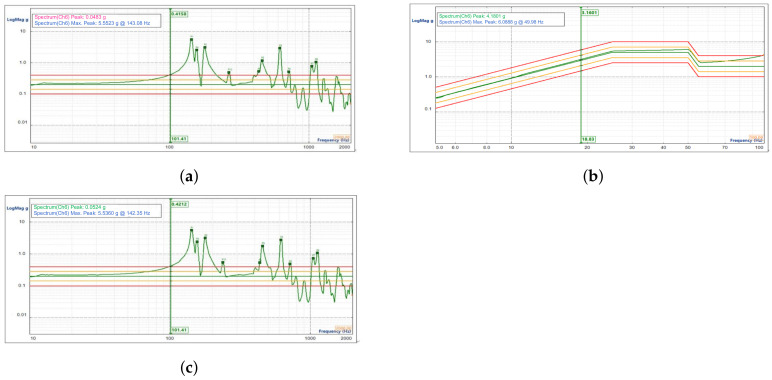
Results of the sinusoidal vibration test on the focusing reflector assembly in the Y direction. (**a**) Results of the first sweep in the Y direction. (**b**) Results of the sinusoidal vibration frequency response in the Y direction. (**c**) Results of the second sweep in the Y direction.

**Figure 14 sensors-23-04610-f014:**
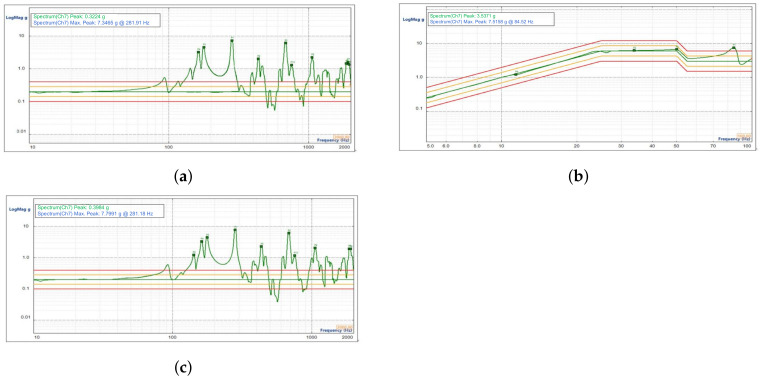
Results of the sinusoidal vibration test on the focusing reflector assembly in the Z direction. (**a**) Results of the first sweep in the Z direction. (**b**) Results of the sinusoidal vibration frequency response in the Z direction. (**c**) Results of the second sweep in the Z direction.

**Table 1 sensors-23-04610-t001:** The physical properties of the selected materials.

						Coefficient of
					Thermal	Thermal
		Density	Y’modulus	Poisson’s	Conductivity	Expansion
Part	Material	($g\cdot {\mathrm{cm}}^{-3}$)	(GPa)	Ratio	($W\cdot m^{-1}\cdot K^{-1}$)	($10^{-6}\cdot K^{-1}$)
mirror	SiC	3.2	330	0.25	185	2.5
insert	4J36	8.9	141	0.3	13.9	2.3
support	TC17	4.4	114	0.3	8.2	9.0
baseplate	AlSiC	3.0	180	0.3	152	8.0
flexible structure	TC4	4.4	109	0.34	8.8	9.1
connectors	Al	2.7	70	0.33	220	23.9

**Table 2 sensors-23-04610-t002:** Piezo actuator technical data from the user manual.

Piezo Actuator Technical Data		
Active axes		X
Displacement		20 mm
Integrated sensor		N-216.2A1: linear encoder
Open-loop resolution		0.03 nm
Close-loop resolution		5 nm
Drive force (active), Fp		600 N
Holding force (passive), Fh		800 N
Lateral force, F1		20 N
Operational temperature range		−40 to 80 ℃
Material		Aluminum, stainless steel
Mass		1250 g

**Table 3 sensors-23-04610-t003:** Simulation modes and test modes.

Modes	Mode 1	Mode 2	Mode 3	Mode 4
Simulation	299 Hz	924 Hz	945 Hz	1180 Hz
Test	285 Hz	910 Hz	935 Hz	1190 Hz

**Table 4 sensors-23-04610-t004:** Results of surface figure and rigid displacements.

Conditions	PV (nm)	RMS (nm)	*d*x (μm)	*d*y (μm)	*d*z (μm)	*r*x (”)	*r*y (”)
Gravity X	19.274	2.381	−14.326	0.01	−0.003	−0.006	2.497
Gravity Y	29.759	6.447	0.005	7.941	0.51	4.068	−0.002
Gravity Z	51.325	8.5125	−0.0005	−13.907	17.016	3.451	−0.003
Temperature variation of 4 °C	42.526	9.043	0.001	1.005	0.268	3.77	0.012
Forced displacement of 0.03 mm	40.648	7.508	0.016	13.704	2.551	43.977	0.000

**Table 5 sensors-23-04610-t005:** Results of the modal analysis of mirror assembly.

Order	F/Hz	Mode of Vibration
1	121.50	Around Y axis direction
2	154.08	Around X axis direction
3	154.25	Around Z axis direction

**Table 6 sensors-23-04610-t006:** Vibration conditions in the frequency range of 5–100 Hz in three directions of X, Y, and Z.

Direction	Frequency			
	5–25/Hz	25–50/Hz	50–55/Hz	55–100/Hz
X	5 mm–5 g	5 g	5 g–2 g	2 g
Y	5 mm–5 g	5 g	5 g–2 g	2 g
Z	5 mm–6 g	6 g	6 g–2 g	2 g

## Data Availability

The data presented in this study are available in article.

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
