# Peer review of "Design of a Focusing Mechanism Actuated by Piezoelectric Ceramics for TMA Telescope"

_sensors, 2023, doi:10.3390/s23104610_

Round 1
Reviewer 1 Report
This article talks about the use of piezoelectric ceramics to create a focusing mechanism for telescopes, specifically the TMA Telescope. The mechanism is designed to be precise and efficient, allowing for better image quality and accuracy. The focusing mechanism designed in this article improves image quality and accuracy for the TMA Telescope by compensating for defocus and line-of-sight jitter error caused by changes in the optical interval and tilt angle. The mechanism has a three-degrees-of-freedom adjustment capability, which allows for adjusting the displacement and rotation of the mirror reflector. This compensation ensures that the imaging quality of the telescope in orbit is maintained. Additionally, the piezoelectric ceramic actuator provides a sufficient driving force and focus stroke, achieving micron-level precision without any gap, movable motion pair, friction, or transmission error.
The article provides complete design, simulation, and experimental data of the flexible structure, it can be considered for publication after revising some minor errors:
1. In line 185 'shown in Figure ??a and Table 2' should be 'shown in Figure 5a and Table 2'.
2. In line 259, 'vibration modes of the flexible structure are shown in Figure ??' should be 'Figure 7'.
3. Figure 9 should have legends for each curve, and please enlarge the font size for X Y axis units to make them more clear. Please polish the Figure 9, it looks very messy now.
4. Please enlarge the legend of Figure 12 (c), Figure 13 (a) and (c), Figure 14 (a), (b), and (c).
Reviewer 2 Report
Comments and Suggestions for Authors are provided in the attached PDF.

The manuscript needs a revision to ameliorate the English style and phrasing.
Reviewer 3 Report
This work discusses the mechanical design of a focusing mechanism for off-axis TMA telescopes, using a folding mirror driven by a piezoelectric ceramic actuator. The actuator components feature a flexible structure that works together with a piezoelectric actuator, providing satisfactory motion range for focusing and simultaneously preventing damage during launch or operation. The authors investigate the vibrational response of the flexible structure, successfully showing that the first four vibrational modes correspond between simulation and experimental results. The authors also examine the reflector’s surface figure under different operating conditions and confirm that it meets the necessary requirements. Finally, vibrational tests were performed on the complete assembly, demonstrating that it meets the required vibrational stiffness criteria.
-Throughout the text, the authors claim various tests were performed to optimize parameters. The tests are not described, and the results are not provided. The following are examples:
“After a thermo-optical analysis, the focusing mechanism is required to have the 132 ability of a 2 mm translation and a 0.03 mm tilt.” What analysis?
“A static analysis and finite element modal analysis of the structure were completed. At the same time, the hammer test was carried out on the processed model. The results show that the finite element model is consistent with the processed model.” Refer to the figures 7-9.
Regarding the design of the flexible ring, “According to the structure size and processing technology, it was determined that n = 4. The variables that need to be optimized are the thickness of the diaphragm flexure T, the number of arms N, and the width ratio R. To analyze the relationship among the changes in the parameters, combinations of parameters, and the stiffness of the structure in the axial and radial directions, the full factorial design accurately evaluated all combinations of all factors at all levels. After the calculations, the optimal dimensions of the flexible structure in the above design were determined to be N = 9, T = 0.5, and R = 1/1.” What specific calculations took place?
-Several improvements to the figures should be made.
Figure 3 should have labels for the components.
Figures 3 and 4 have the same caption.
Figure 5 uses a picture from the Physik Instrumente website. An original schematic is more appropriate.
Figure 6b is described as the optimized structure, but is not the optimized structure that is calculated/tested in the following figures.
Difficult to interpret figure 9, would help to process the raw data and make labels for the axes more visible.
Table 3 needs units.
Figure 10 needs units.
It could be made clearer what is being shown in figure 11, is this the completed assembly? Maybe labels would help.
Figure 12 captions says “Schematic of incident light and reflected light when the mirror is translated and rotated” I think this is misplaced text.
Figures 12-14 are raw data missing y axes labels.
-Formatting issues with "Figure ??" appearing several times.
Other than these concerns, the manuscript is well-written and will be suitable for publication after revisions.
Round 2
Reviewer 2 Report
The authors have considered all my comments and they have ameliorated the manuscript.
For final editing, some details of the manuscript can be ameliorated in terms of English Language.